# Repetitive Transcranial Magnetic Stimulation as an Add-On Treatment for Cognitive Impairment in Alzheimer’s Disease and Its Impact on Self-Rated Quality of Life and Caregiver’s Burden

**DOI:** 10.3390/brainsci11060740

**Published:** 2021-06-03

**Authors:** Juliana Teti Mayer, Caroline Masse, Gilles Chopard, Magali Nicolier, Matthieu Bereau, Eloi Magnin, Julie Monnin, Gregory Tio, Emmanuel Haffen, Pierre Vandel, Djamila Bennabi

**Affiliations:** 1Laboratoire de Recherches Intégratives en Neurosciences et Psychologie Cognitive, Université Bourgogne Franche-Comté, 25000 Besançon, France; cmasse@chu-besancon.fr (C.M.); gilleschopard@gmail.com (G.C.); mnicolier@chu-besancon.fr (M.N.); mbereau@chu-besancon.fr (M.B.); eloi.magnin@univ-fcomte.fr (E.M.); jmonnin@chu-besancon.fr (J.M.); emmanuel.haffen@univ-fcomte.fr (E.H.); pierre.vandel@univ-fcomte.fr (P.V.); djamila.bennabi@univ-fcomte.fr (D.B.); 2Service de Psychiatrie de l’Adulte, Centre Hospitalier Universitaire de Besançon, CEDEX, 25030 Besançon, France; gtio@chu-besancon.fr; 3Service de Neurologie, Centre Hospitalier Universitaire de Besançon, CEDEX, 25030 Besançon, France; 4Centre Mémoire Ressources et Recherche, Centre Hospitalier Universitaire de Besançon, CEDEX, 25030 Besançon, France; 5Centre d’Investigation Clinique, INSERM CIC 1431, Centre Hospitalier Universitaire de Besançon, CEDEX, 25030 Besançon, France; 6Centre Expert Dépression Résistante FondaMental, Centre Hospitalier Universitaire de Besançon, CEDEX, 25030 Besançon, France

**Keywords:** rTMS (repetitive transcranial magnetic stimulation), Alzheimer’s disease, dorsolateral prefrontal cortex, cognitive impairment, quality of life, memory

## Abstract

Alzheimer’s disease (AD) is associated with progressive memory loss and decline in executive functions, as well as neuropsychiatric symptoms. Patients usually consider quality of life (QoL) and mood as more important for their health status than disease-specific physical and mental symptoms. In this open-label uncontrolled trial, 12 subjects diagnosed with AD underwent 10 sessions of repetitive transcranial magnetic stimulation (rTMS) over the left dorsolateral prefrontal cortex (10 Hz, 20 min, 2000 pulses/day, 110% MT). Outcomes were measured before and 30 days after treatment. Our primary objective was to test the efficacy of rTMS as an add-on treatment for AD on the global cognitive function, assessed through the Mini-Mental State Examination (MMSE) and the Mattis Dementia Rating Scale (MDRS). As secondary objectives, the detailed effect on cognitive functions, depression and anxiety symptoms, QoL, and functionality in daily life activities were evaluated, as well as correlations between QoL and cognition, depression and anxiety scores. The treatment significantly enhanced semantic memory and reduced anxiety. Improvement of these features in AD could become an important target for treatment strategies. Although limited by its design, this trial may contribute with another perspective on the analysis and the impact of rTMS on AD.

## 1. Introduction

Alzheimer’s disease (AD) is the most common type of dementia among the elderly and one of the heaviest burdens on the public health system. People with AD may suffer from progressive memory loss, aphasia, and declines in executive functions as well as neuropsychiatric symptoms including depression, hallucinations, and apathy, which seriously impact the activities of daily living and quality of life (QoL) of patients and, consequently, their caregivers [1,2]. Depression and cognitive dysfunction are interdependent and associated with various adverse consequences, such as poor treatment compliance, loss of independence, and even mortality [3,4,5]. Patients with brain disorders often perceive QoL and depressive mood as more important for their health status than disease-specific physical and mental symptoms. Therefore, improvement of these common features should become an important target in treatment of AD.

Despite advances in the pharmacopeia, there is a lack of effective treatment to lessen cognitive symptoms and modify the progression of the disease. Current medications such as acetylcholinesterase inhibitors (AChEI) and N-methyl-D-aspartic (NMDA) glutamate receptors have demonstrated a symptomatic effect on certain cognitive and non-cognitive symptoms of AD in the short term: six to twelve months in most studies [6]. With diagnostic tools for AD becoming increasingly sophisticated, the pathology is identified at earlier stages than before, and therapies that may limit the progression of the illness and the cognitive loss associated with it are needed.

Among nonpharmacological interventions, repetitive transcranial magnetic stimulation (rTMS), with its potential to modify brain activity in targeted brain areas and related neural networks, seems to be a promising approach. This technique uses brief duration, rapidly alternating, or pulsed magnetic fields, delivered through an electromagnetic coil placed on the surface of the scalp to modulate the activity of cortical neurons underneath the application site. Its application at low frequencies (≤1 Hz) tends to suppress cortical excitability, whereas high frequency (HF) stimulation (>5 Hz) tends to enhance or facilitate cortical excitability through long-term potentiation (LTP)-like effects. When used repeatedly, such modulatory effect of cortical neural activity can persist beyond the period of stimulation, justifying the therapeutic use of rTMS. The effects of rTMS can depend on stimulation parameters that affect the electromagnetic field generated, such as coil shape, waveform, and pulses, session protocol, and on interindividual and disease-related specificities that affect the clinical response [7,8].

In AD, there are encouraging results for the use of HF-rTMS (10–20 Hz) unilaterally, over the left dorsolateral prefrontal cortex (left dlPFC), or bilaterally [9]. Improvements have been found in functional performance, global cognition—in domains such as episodic memory, psychomotor speed, and language skills—as well as in depressive symptoms [9,10,11,12]. Another protocol applied stimulation over several brain regions bilaterally, concurrently with cognitive training for 6 weeks, finding medium to large effect size improvements (0.4–0.7) in neuropsychological, clinical and functional assessments up to 4.5 months [13]. In spite of the accumulating evidence of positive outcomes brought by rTMS in the treatment of AD-related cognitive impairment, issues about the exact therapeutic effects and their impact on quality of life and daily functioning of the patients require further research.

We have therefore conducted an open-label uncontrolled pilot study to evaluate whether 10 HF-rTMS applied to the left dlPFC as an add-on treatment could positively affect cognition, QoL, functional ability, and psychiatric symptoms of AD patients. The primary objective was to analyze changes in global cognitive impairment one month after rTMS sessions. As secondary outcome measures, we also investigated the neuromodulatory effects on specific cognitive functions (attention/processing speed, executive function, episodic memory, and language/semantic functioning), QoL, functionality in daily activities (from both the patient’s and the caregiver’s perspectives), and depressive and anxiety symptoms. Lastly, correlation between these variables, QoL, and autonomy were also explored, both at baseline and post-treatment assessments.

## 2. Materials and Methods

### 2.1. Participants

Twelve patients diagnosed with AD—according to the National Institute of Neurological and Communicative Disorders and Stroke, Alzheimer’s Disease and Related Disorders Association (NINCDS-ADRDA) criteria [14]—were recruited from the Memory Center of Research and Resources of the University Hospital of Besançon, France. Besides the (i) AD diagnosis as inclusion criteria, patients were also required to (ii) score ≤ 2 on the Clinical Dementia Rating scale, and (iii) be treated with IAChE during the 3 months preceding stimulations and the follow-up period. Patients were excluded in case of (i) psychiatric comorbidity, (ii) contraindication for brain magnetic resonance imaging or TMS, (iii) severe white matter lesions (>Fazekas grade 2), or (iv) severe cortical atrophy. Each participant (or primary caregiver) signed an informed consent to participate in the study. The research protocol was approved by the Committee of Protection of Persons EST II (CPP-EST-II, Saint-Jacques Hospital, Besançon, France) and was conducted in accordance with the principles of the Declaration of Helsinski.

### 2.2. Study Design

In this open-label uncontrolled pilot trial, all subjects underwent the same procedure. Each patient received 10 sessions of rTMS, delivered twice a day over a period of 5 days, as an add-on treatment in their follow-up. A Magstim Super Rapid2 (Magstim Company Ltd., Whitland, Wales, UK) with an air-cooling figure-of-eight coil was used. The rTMS was administered at 10 Hz during 5 s, with 25 s between trains (parameters previously associated with a positive impact of rTMS as an adjunctive treatment on cognition in AD [12]), and at 110% motor threshold (MT) over the left dlPFC per 20 min session (2000 stimuli per day). The cortical MT was defined as the minimum intensity that produced a slight contraction of the contralateral muscle abductor pollicis brevis. The coil was placed tangentially to the head in a posterior–anterior orientation. Position was kept constant by a static mechanical support. Directly below the center of the coil, the placement to stimulate the left dlPFC was 5 cm anterior to the hand motor area, in a parasagittal line. Outcome measures were evaluated at baseline (D0) and 30 days after the end of the treatment (M1). Tolerability and side effects were assessed both verbally, during and after sessions, and through a Visual Analogue Scale (VAS) applied by the end of sessions. The VAS assessed the patient’s perception on pain, mood, fatigue, and motivation. Cutoff points for side effects were set as follows, from the left to the right endpoint: absent, mild, moderate, and severe corresponded to the first, second, third, and fourth quarters of the scale, respectively.

This study was conducted in preparation to a randomized double blind controlled trial (in case of favorable results), thus no-sham stimulation was included in this phase. The outcomes were tested one month following rTMS treatment since exploration of long-term effects is more interesting considering a clinical perspective and the intended influence on the patients’ quality of life and daily functioning.

### 2.3. Outcome Measures

#### 2.3.1. Cognitive Assessment

An experienced neuropsychologist conducted the cognitive tests at baseline (D0) and 30 days after the end of the treatment (M1). Our primary outcome measure was the patient’s global cognition, rated through the Mini-Mental State Examination (MMSE) and the Mattis Dementia Rating Scale (MDRS) [15]. Additionally, as secondary measures, the following cognitive domains were assessed: (i) Attention and Processing speed: Crossing-Off Test (COT) [16], Trail Making Test-Part A (TMT-A) [17], and MDRS attention subscale; (ii) Executive function: TMT Part B-Part A (TMT B-A), Isaacs Set Test (IST) [18], and MDRS initiation/perseveration (I/P) subscale; (iii) Episodic memory: Delayed Matching-to-Sample task (DMS-48) [19] and MDRS memory subscale; and (iv) Language and semantic functioning: Oral picture naming test (DO-30) [20] and MDRS conceptualization subscale.

#### 2.3.2. Functional Ability and Quality of Life Assessment

Patients’ functional ability and autonomy in activities of daily living were also part of the secondary outcomes of this study, rated at baseline (D0) and 30 days after the end of the treatment (M1). These variables were evaluated through Katz’s Activities of Daily Living scale (ADL) [21], and Lawton’s Instrumental Activities of Daily Living scale (IADL) [22]. Both scales were completed in accordance with the caregivers’ statements. Quality of life was assessed through the QoL in Alzheimer’s Disease scale (QoL-AD) [23,24].

#### 2.3.3. Psychiatric Assessment

The last secondary measures—depression and anxiety symptoms—were as well evaluated at baseline (D0) and 30 days after the end of the treatment (M1) by a trained psychiatrist. Rating scales of depression have included the 21-items Hamilton Depression Rating Scale (HDRS-21) [25] and the self-rated Beck Depression Inventory (BDI) [26]. Anxiety was measured with Spielberger’s State-Trait Anxiety Inventory, forms YA and YB (STAI-YA and YB) [27].

### 2.4. Statistical Analysis

Analyses were performed using SAS 9.4 T5 Level MM3. Given the sample size, Wilcoxon sign rank test (nonparametric test) was performed for all variable comparisons. Alpha value for significance was set at 0.05 and Bonferroni’s correction was applied to significance level according to the number of comparisons performed in each domain (among primary and secondary outcome measures). Spearman’s rho was used to explore correlations with significance level set at 0.05.

## 3. Results

### 3.1. Demographic and Baseline Characteristics

Participants’ demographic and baseline characteristics are summarized on Table 1. Data were missing at random for some subjects, so the number of subjects whose data were considered for analysis is detailed in the “Subjects (*n*)” column. Our sample was composed of nine women and three men, with a mean age of 70.2 (SD ± 6.5) years old. Seven participants (58%) had a secondary education degree, three participants completed higher education, one had a high school degree, and one finished elementary education. All of them were retired from their former jobs. Nine patients were in monotherapy (rivastigmine = 5; galantamine = 2; memantine = 1; donepezil = 1), while three were in combined therapy (memantine and: rivastigmine = 2; donepezil = 1).

Before starting HF-rTMS treatment, we have observed that some baseline measures were significantly correlated with the participants’ QoL and functional ability assessment (see Table 2). Their MDRS attention and memory subscales were directly correlated with their QoL-AD score (*r* = 0.812, *p* = 0.01 and *r* = 0.686, *p* = 0.04, respectively). Regarding the autonomy on daily activities (IADL score), there was a positive correlation with the MMSE score (*r* = 0.680; *p* = 0.03), the global MDRS score (*r* = 0.735; *p* = 0.01), the MDRS memory subscale (*r* = 0.855; *p* < 0.01), the IST (*r* = 0.784; *p* = 0.01), and the DO-30 (*r* = 0.639; *p* = 0.05). On the other hand, the same IADL score was negatively correlated with the COT score (*r* = −0.830; *p* < 0.01). The other daily functioning measure (ADL score) was as well positively correlated with the IST (*r* = 0.719; *p* = 0.02) and negatively correlated with the COT performance (*r* = −0.676; *p* = 0.03).

### 3.2. Effect of 10 HF-rTMS Sessions

All patients received the sessions at 10 Hz with the intensity of 110% of their MT. The stimulation sessions had no significant impact on participants’ global cognition, our primary outcome measure (corrected significance level: *p* < 0.02), assessed through the MMSE (*p* = 0.30) and the global MDRS score (*p* = 0.10). When it comes to our secondary measures, Bonferroni’s correction was used according to the number of applied measures per assessed domain (see the first column on Table 3 and Table 4). A significant effect on the cognitive function of semantic memory was observed through the performance on the MDRS conceptualization subscale (*p* = 0.01; Figure 1a). The effect on the visual recognition memory, assessed through the DMS-48, was marginally significant (*p* = 0.04). Regarding the psychiatric evaluations, a significant impact of treatment was observed on trait anxiety, as assessed through the STAI-YB (*p* = 0.01; Figure 1b). No other significant effects were detected. Data for all outcome measures are displayed on Table 3 and Table 4, under their respective categories.

### 3.3. Correlations between Response to Treatment and Quality of Life

Since there was no rating change on functional ability (ADL and IADL) from baseline to 1-month post-treatment in our population (Table 3), we have only explored the correlations between the response to treatment of the different outcome measures and the effect on QoL-AD scores. They were negatively correlated with trait anxiety (STAI-YB) scores (*r* = −0.736; *p* = 0.04) and self-rated depression (BDI) scores (*r* = –0.768; *p* = 0.03), as displayed on Table 5.

### 3.4. Side Effects

Neither during sessions nor after treatment did participants report any side effects. All treatment sessions were well tolerated, and all participants scored in the first quarter of the scale (absence of side effects).

## 4. Discussion

This study aimed to investigate the clinical efficacy of HF-rTMS as an add-on treatment in improving cognition, QoL, and functional ability in daily life activities—from the patient’s and the caregiver’s perspective—as well as depression and anxiety rates of patients diagnosed with AD. Sessions were well tolerated and no adverse effects were reported by the participants. After treatment, a significant improvement in the patients’ semantic memory was observed in the 1-month follow-up evaluation. A significant reduction on the trait anxiety scores was as well detected, and this effect was directly correlated with an improvement on QoL. Reduction of self-reported depressive symptoms was equally associated with improvement on QoL, although the impact of treatment was marginally significant.

The World Health Organization (WHO) defines QoL as “an individual’s perception of their position in life in the context of the culture and value systems in which they live and in relation to their goals, expectations, standards and concerns. It is a broad ranging concept incorporating in a complex way the person’s physical health, psychological state, level of independence, social relationships, personal beliefs and their relationship to salient features of the environment” [28]. In line with this concept, the baseline data from our population suggested that the patients’ global cognition, as well as cognitive functions from all studied categories, directly impacted their QoL and autonomy in daily life activities. It is worthy noticing that the better the QoL reported by the patients, the higher they performed on the attention and memory MDRS subscales. These findings could reinforce our starting point, that cognition might be an important therapeutic target in AD in order to positively influence the patients’ perception of their QoL and the caregiver’s burden.

One month after rTMS treatment, we have found a significant improvement in the conceptualization subscale of the MDRS, which consists of tasks such as identifying the similarities between pairs of objects, identifying non-members of semantic categories, or identifying similarities and differences among simple geometric figures. The left dlPFC is mainly involved in semantic retrieval [29], and it could be assumed that rTMS sessions targeting this region contribute to enhance the MDRS conceptualization subscale by strengthening the semantic-based treatment. Our data also showed an improvement marginally significant on participants’ visual recognition memory performance. Impaired visual memory recognition reported in 30 AD patients, and detected through the DMS-48, could be particularly related to medial temporal and perirhinal dysfunction [19], the first regions affected in AD according to the autopsy of 83 brains of non-demented and demented individuals [30].

Considering the practice (test–retest) effect is an important aspect when investigating treatment response through cognitive assessment [31]. Scharfen and colleagues [32] recently performed a meta-analysis and observed that, for non-clinical samples, a mere repetition of a cognitive ability test could result in an improvement of a third of a standard deviation (of the baseline model), which would reach stability after the third-fourth administration. Calamia et al. [33] had described a similar effect, of almost a quarter of a standard deviation in a one-year interval. However, they detected that this effect was smaller in clinical populations and, according to their findings, AD patients would be expected to show an average decrease of 27.5% of a standard deviation if retested in a one-year period [33]. Additionally, other studies have pointed out that practice effects are largely absent in patients with dementia [34,35,36], even for those with mild AD for short test–retest intervals [37,38]. Nevertheless, given that in the present study the retest interval is of one month, questions could still be raised on the influence of a practice effect regarding the results. This effect should be taken into account, adapted to the context of AD patients, in future studies with larger samples and long-term controlled follow-ups, as a careless analysis of cognitive outcomes could indeed lead to misinterpretation of the patients’ performances.

According to a recent review, positive cognitive effects of rTMS were often reported in mild AD while treatment failed to improve more advanced cases, suggesting that the efficacy may depend on the stage of the disease [7]. Furthermore, in a meta-analysis by Chou and colleagues [11], the included studies suggested that the significant effect of HF-rTMS over the left dlPFC of patients with AD or mild cognitive impairment would be the enhancement of memory function, and that the effects of 5–30 consecutive sessions could last for 4–12 weeks. Their results pointed out that memory improvement could as well be obtained following application of low-frequency rTMS over the right dlPFC [11]. The left dlPFC seems to be preferentially involved in working memory and executive function, according to a literature review [39] and a trial comparing the performance of 30 mild AD patients and 31 matched controls in an antisaccade task [40]. In addition, a significant improvement in recognition memory has been observed after comparing sham to active neuromodulation over the left dlPFC of 10 AD patients [41], while the familiarity component of recognition memory was found to be affected by frontal lobe lesions, due to tumor or stroke, on 24 patients [42]. It could be hypothesized that rTMS sessions over the left dlPFC could improve the encoding phase [41] in the context of the DMS-48 memory assessment, as well as the ability to distinguish between targets and distractor items [42].

Finally, the HF-rTMS treatment significantly improved the trait anxiety (STAI-YB) scores of our sample, with a marginally significant impact on self-reported depressive symptoms (BDI). Correlation analysis on the impact of treatment showed that the decrease of depression and trait anxiety scores was associated with an improvement on QoL. The impact of rTMS on anxiety symptoms lacks further evidence on the literature [43], especially in the context of AD. Our result could thus be seen as promising, suggesting a stable, long-term reduction of anxiety. In addition, this reasoning is also in line with the aforementioned definition of the WHO on quality of life. Although our subjects did not display a significant improvement in their QoL scores, the trait anxiety might represent an important therapeutic target in order to contribute to a better QoL. However, further investigation is still required. On the other hand, the effect on depressed mood, although marginally significant, is well reported in the literature [44]. In this way, further studies should consider including AD patients with clinically important depressive and anxiety symptoms, and perform a more detailed assessment of QoL (e.g., one of the WHO Quality of Life instruments), as means to investigate the therapeutic potential of rTMS as an add-on treatment, and the possible long-term impact in the QoL of this population.

This study has important limitations, which are its relatively small sample size and a lack of control group. These aspects should be taken into account when interpreting our data as well as the possible influence of a test–retest effect, as mentioned above or even fluctuations in the patients’ scores. On the other hand, our trial is strengthened by well-defined inclusion/exclusion criteria, use of previously validated instruments only, and statistical correction for multiple comparisons. The results and the discussion might contribute to guide further research towards improvement of AD patients’ QoL and autonomy.

## 5. Conclusions

Given the current limitations of pharmacological treatments for AD, non-pharmacological interventions such as rTMS have been increasingly explored. We have conducted an open-label trial and described a potential positive impact of 10 well-tolerated HF-rTMS sessions over the left dlPFC on the semantic memory of AD patients and on (trait) anxiety. In addition, this reduction on anxiety was significantly associated with an improvement on QoL scores. However, no impact on patients’ autonomy could be reported in our population. Randomized controlled trials with larger samples and a long-term follow-up, combining stimulation interventions to the patient’s pharmacological treatment, should be encouraged as means to further explored rTMS potential long-term benefits for AD patients and their caregivers.

## Figures and Tables

**Figure 1 brainsci-11-00740-f001:**
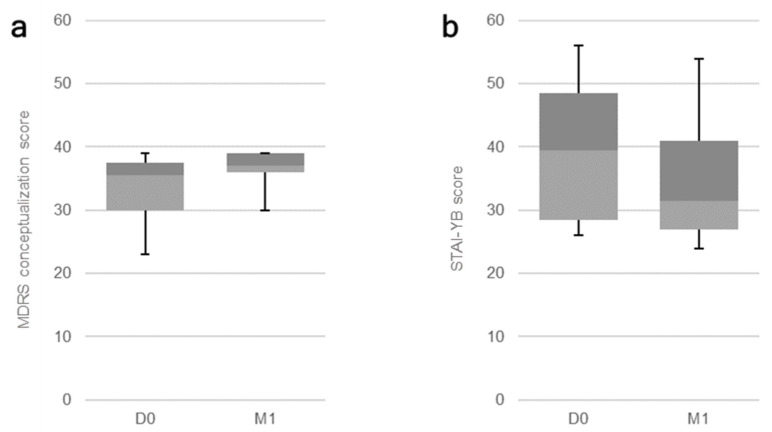
Boxplot for significant effects of rTMS sessions on AD patients: Baseline (D0) and post-treatment (M1) results for the (**a**) MDRS conceptualization subscale (*p* = 0.01) and the (**b**) STAI-YB score for trait anxiety (*p* = 0.01).

**Table 1 brainsci-11-00740-t001:** Mean values, standard deviation (SD) and range scores (Min-Max) for demographic and baseline measures.

	Subjects (*n*)	Mean	±SD	Min	Max
Age	12	70.2	6.5	62	84
MMSE	12	19.1	5.5	11	27
MDRS	12	113.8	19.1	78	136
MDRS Attention	12	34.2	1.9	32	36
MDRS I/P	12	26.2	10.5	8	37
MDRS Memory	12	14.2	4.0	8	23
MDRS Conceptualization	12	33.7	5.2	23	39
COT	12	73.7	32.9	42	130
TMT-A	11	73.0	39.0	24	150
TMT B-A	8	193.7	187.2	42	638
IST	12	21.4	10.8	3	37
DMS-48	12	81.2	11.9	64	98
DO-30	12	25.1	6.7	14	30
QoL-AD	9	33.2	4.0	30	43
ADL	10	5.7	0.5	4.5	6.0
IADL	10	5.3	2.4	2.0	8.0
BDI	12	6.8	4.5	0	15
HDRS-21	12	7.5	4.9	0	17
STAI-YA	12	36.3	11.7	21	59
STAI-YB	12	39.7	11.0	26	56

ADL Activities of Daily Living scale, BDI Beck Depression Inventory, COT Crossing-Off Test, DMS-48 Delayed Matching-to-Sample task, DO-30 Oral picture naming test, HDRS-21 21-items Hamilton Depression Rating Scale, IADL Instrumental Activities of Daily Living scale, IST Isaac’s Set Test, MDRS Mattis Dementia Rating Scale, MDRS I/P initiation/perseveration subscale, MMSE Mini-Mental State Examination, QoL-AD Quality of Life in Alzheimer’s Disease scale, STAI-YA and YB State-Trait Anxiety Inventory forms YA and YB, TMT-A Trail Making Test-Part A, TMT B-A Trail Making Test Part B-Part A.

**Table 2 brainsci-11-00740-t002:** Spearman’s Rho for baseline measures’ (D0) correlations with quality of life and functional ability.

D0	QoL-AD	ADL	IADL
MMSE	0.651	0.260	0.680 *
MDRS	0.582	0.517	0.735 *
MDRS Attention	0.812 *	0.072	0.396
MDRS I/P	0.241	0.462	0.557
MDRS Memory	0.686 *	0.293	0.855 *
MDRS Conceptualization	0.308	0.590	0.604
COT	−0.381	−0.676 *	−0.830 *
TMT-A	−0.096	−0.598	−0.809
TMT B-A	−0.429	−−− ^†^	−0.334
IST	0.051	0.719 *	0.784 *
DMS-48	−0.216	0.203	0.351
DO-30	0.106	0.593	0.639 *
BDI	−0.359	−0.080	−0.434
HDRS-21	0.140	0.011	−0.034
STAI-YA	−0.165	0.451	−0.071
STAI-YB	−0.532	0.360	−0.043

ADL Activities of Daily Living scale, BDI Beck Depression Inventory, COT Crossing-Off Test, DMS-48 Delayed Matching-to-Sample task, DO-30 Oral picture naming test, HDRS-21 21-items Hamilton Depression Rating Scale, IADL Instrumental Activities of Daily Living scale, IST Isaac’s Set Test, MDRS Mattis Dementia Rating Scale, MDRS I/P initiation/perseveration subscale, MMSE Mini-Mental State Examination, QoL-AD Quality of Life in Alzheimer’s Disease scale, STAI-YA and YB State-Trait Anxiety Inventory forms YA and YB, TMT-A Trail Making Test-Part A, TMT B-A Trail Making Test Part B-Part A. * *p* < 0.05. ^†^ No correlation could be established between TMT B-A and ADL scores; all participants with a TMT B-A score (8 subjects) had the same ADL score.

**Table 3 brainsci-11-00740-t003:** Median values and quartiles [Q1;Q3] for pre- (D0) and 1-month post-10 rTMS sessions (M1) measures with *p*-values.

Measure (Corrected Significance Level)	Median [Q1;Q3]		*p*-Value
	D0	M1	
Global cognition (*p* < 0.02)			
MMSE	20 [14;23.5]	20.5 [16;24]	0.30
MDRS	119.5 [100.5;127.5]	123 [118;127]	0.10
Attention/Processing speed (*p* < 0.01)			
COT	59 [52;96.5]	58 [46;87]	0.92
TMT-A	67 [38;102]	48 [35;70]	0.12
MDRS Attention	35 [32;36]	36 [35;36]	0.25
Executive function (*p* < 0.01)			
TMT B-A	134.5 [99.5;201]	165.5 [91;440]	0.38
IST	23.5 [13.5;28.5]	21 [16;30]	0.79
MDRS I/P	31 [19;33]	30 [29;32]	0.52
Episodic memory (*p* < 0.02)			
DMS-48	81 [70;93.5]	87.5 [76;97]	0.04
MDRS Memory	14.5 [11.5;16]	14 [11;17]	0.65
Language/Semantic functioning (*p* < 0.02)			
DO-30	29 [18.5;30]	28.5 [19;30]	0.78
MDRS Conceptualization	35.5 [30;37.5]	37 [36;39]	0.01 *
Functional ability (*p* < 0.02)			
ADL	6 [5.5;6]	6 [5.7;6]	1
IADL	5 [3;8]	6 [3.5;8]	1
Quality of life (*p* < 0.05)			
QoL-AD	32 [31;34]	32 [30;34]	0.28

ADL Activities of Daily Living scale, COT Crossing-Off Test, DMS-48 Delayed Matching-to-Sample task, DO-30 Oral picture naming test, IADL Instrumental Activities of Daily Living scale, IST Isaac’s Set Test, MDRS Mattis Dementia Rating Scale, MDRS I/P initiation/perseveration subscale, MMSE Mini-Mental State Examination, QoL-AD Quality of Life in Alzheimer’s Disease scale, TMT-A Trail Making Test-Part A, TMT B-A Trail Making Test Part B-Part A. * Significant effect.

**Table 4 brainsci-11-00740-t004:** Median values and quartiles [Q1;Q3] for pre- (D0) and 1-month post-10 rTMS sessions (M1) measures with *p*-values.

Measure (Corrected Significance Level)	Median [Q1;Q3]		*p*-Value
	D0	M1	
Depression (*p* < 0.02)			
BDI	7 [4;9.5]	3.5 [0.5;6.5]	0.03
HDRS-21	6.5 [4.5;11]	5.5 [2.5;8]	0.05
Anxiety (*p* < 0.02)			
STAI-YA	37 [26;43.5]	33.5 [27;39.5]	0.31
STAI-YB	39.5 [28.5;48.5]	31.5 [27;41]	0.01 *

BDI Beck Depression Inventory, HDRS-21 21-items Hamilton Depression Rating Scale, STAI-YA and YB State-Trait Anxiety Inventory forms YA and YB. * Significant effect.

**Table 5 brainsci-11-00740-t005:** Spearman’s Rho for the correlation between the impact of treatment (M1-D0) on variables and on quality of life.

M1-D0	QoL-AD
MMSE	0.413
MDRS	0.706
MDRS Attention	0.633
MDRS I/P	0.545
MDRS Memory	0.695
MDRS Conceptualization	−0.222
COT	−0.691
TMT-A	−0.109
TMT B-A	0.200
IST	0.701
DMS-48	0.390
DO-30	0.241
BDI	−0.768 *
HDRS-21	−0.412
STAI-YA	−0.337
STAI-YB	−0.736 *

BDI Beck Depression Inventory, COT Crossing Off Test, DMS-48 Delayed Matching-to-Sample task, DO-30 Oral picture naming test, HDRS-21 21-items Hamilton Depression Rating Scale, IST Isaac’s Set Test, MDRS Mattis Dementia Rating Scale, MDRS I/P initiation/perseveration subscale, MMSE MiniMental State Examination, QoL-AD Quality of Life in Alzheimer’s Disease scale, STAI-YA and YB State-Trait Anxiety Inventory forms YA and YB, TMT-A Trail Making Test-Part A, TMT B-A Trail Making Test Part B-Part A. * *p* < 0.05.

## Data Availability

The datasets generated and analyzed during the current study are available from the corresponding author on reasonable request.

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
