# Peer review of "Repetitive Transcranial Magnetic Stimulation as an Add-On Treatment for Cognitive Impairment in Alzheimer’s Disease and Its Impact on Self-Rated Quality of Life and Caregiver’s Burden"

_brainsci, 2021, doi:10.3390/brainsci11060740_

Round 1
Reviewer 1 Report
In the absence of a control (sham stimulation) group there is no way to assess the impact of practice in test-taking. The results of the meta-analyses of Scharfen and Calamia are not particularly relevant given the large difference in the test-retest interval (greater test-retest improvement is expected in the present study).
However, the modest-size correlations between change in cognitive performance on certain tests and QoL may represent a promising trend. Given the very small sample size inspection of bivariate scatter plots should be inspected to ensure that these correlations were not inflated by the presence of bivariate outliers.
The significance of this study in comparison to previous ones requires that the sample sizes of previous studies are mentioned in the Discussion (line 285-293).
Given the diagnosis of participants were they all considered as capable of providing informed consent? For instance what was the range of MMSE scores across participants?
Author Response
REVIEWER 1
In the absence of a control (sham stimulation) group there is no way to assess the impact of practice in test-taking. The results of the meta-analyses of Scharfen and Calamia are not particularly relevant given the large difference in the test-retest interval (greater test-retest improvement is expected in the present study).
Thank you for the important remark; we have modified the following paragraph in order to address this matter:
Lines 306-310: “Calamia et al. [35] had described a similar effect, of almost a quarter of a standard devia-tion in a one-year interval. However, they detected that this effect was smaller in clinical populations and, according to their findings, AD patients would be expected to show an average decrease of 27.5% of a standard deviation if retested in a one-year period [35]. Given that in the present study the retest interval is of one month, questions could still be raised on the influence of a practice effect regarding the results. This effect should be taken into account – adapted to the context of AD patients – in future studies with larger sam-ples and long-term follow-ups, as a careless analysis of cognitive outcomes could indeed overlook improvements, if a decline of functions is not considered.”
However, the modest-size correlations between change in cognitive performance on certain tests and QoL may represent a promising trend. Given the very small sample size inspection of bivariate scatter plots should be inspected to ensure that these correlations were not inflated by the presence of bivariate outliers.
Thank you for the remark. Scatter plots of all significant correlations have been added as supplementary material. Please see the attached file.
The significance of this study in comparison to previous ones requires that the sample sizes of previous studies are mentioned in the Discussion (line 285-293).
Thank you for the suggestion, the samples sizes have been added as follows:
Lines 293-302: “The left dlPFC seems to be preferentially involved in working memory and executive func-tion, according to a literature review [30] and a trial comparing the performance of 30 mild AD patients and 31 matched controls in an antisaccade task [31]. In addition, a significant improvement in recognition memory has been observed after comparing sham to active neuromodulation over the left dlPFC of 10 AD patients [32], while the familiarity compo-nent of recognition memory was found to be affected by frontal lobe lesions, due to tumor or stroke, on 24 patients [33]. It could be hypothesized that rTMS sessions over the left dlPFC could improve the encoding phase [32] in the context of the DMS-48 memory as-sessment, as well as the ability to distinguish between targets and distractor items [33].”
Given the diagnosis of participants were they all considered as capable of providing informed consent? For instance what was the range of MMSE scores across participants?
Thank you for the observation, the following sentence has been modified accordingly:
Lines 102-103: “Each participant (or primary caregiver) signed an informed consent to participate in the study.”
In addition, a column was added in Table 1 with range scores for all baseline measures. Concerning the MMSE scores: Min = 11 and Max = 27.

Reviewer 2 Report
The seriousness of Alzheimer's Disease warrants the publication of exploratory studies such as this one. The authors appropriately note the limitations of the study, and ultimately, they call for further research on the topic.
As a former statistics professor, I was pleased to see the use of the Bonferroni correction for the multiple analyses.
A truly minor point-- the over-reliance on acronyms made an otherwise well written paper difficult to read. I kept having to go back and search to find out what each acronym meant. If authors could figure out how to reduce acronym use, it was be wonderful.
Author Response
REVIEWER 2
The seriousness of Alzheimer's Disease warrants the publication of exploratory studies such as this one. The authors appropriately note the limitations of the study, and ultimately, they call for further research on the topic.
As a former statistics professor, I was pleased to see the use of the Bonferroni correction for the multiple analyses.
A truly minor point-- the over-reliance on acronyms made an otherwise well written paper difficult to read. I kept having to go back and search to find out what each acronym meant. If authors could figure out how to reduce acronym use, it was be wonderful.
Thank you for the comments. The authors have replaced some of the acronyms by their measured outcome (e.g. “QoL-AD” by “quality of life”) or added a description of the measure close to the acronyms (e.g. “…improved the trait anxiety (STAI-YB) scores of our sample, with a marginally significant impact on self-reported depressive symptoms (BDI).”) in order to enhance the quality of writing.